# Chemical Composition of *Ducrosia flabellifolia* L. Methanolic Extract and Volatile Oil: ADME Properties, In Vitro and In Silico Screening of Antimicrobial, Antioxidant and Anticancer Activities

**DOI:** 10.3390/metabo13010064

**Published:** 2022-12-31

**Authors:** Mejdi Snoussi, Ramzi Hadj Lajimi, Riadh Badraoui, Mousa Al-Reshidi, Mohammad A. Abdulhakeem, Mitesh Patel, Arif Jamal Siddiqui, Mohd Adnan, Karim Hosni, Vincenzo De Feo, Flavio Polito, Adel Kadri, Emira Noumi

**Affiliations:** 1Department of Biology, College of Science, University of Hail, P.O. Box 2440, Hail 2440, Saudi Arabia; 2Laboratory of Genetics, Biodiversity and Valorization of Bio-Resources (LR11ES41), Higher Institute of Biotechnology of Monastir, University of Monastir, Avenue Tahar Haddad, BP74, Monastir 5000, Tunisia; 3Department of Chemistry, College of Science, University of Hail, P.O. Box 2440, Hail 2440, Saudi Arabia; 4Laboratory of Water, Membranes and Environmental Biotechnologies, Center of Research and Water Technologies, P. B 273, Soliman 8020, Tunisia; 5Section of Histology Cytology, Medicine Faculty of Tunis, University of Tunis El Manar, La Rabta 1007, Road Djebal Lakhdhar, Tunis 1007, Tunisia; 6Molecular Diagnostics and Personalized Therapeutics Unit, University of Hail, P.O. Box 2440, Hail 2440, Saudi Arabia; 7Department of Biotechnology, Parul Institute of Applied Sciences, Centre of Research for Development, Parul University, Vadodara 391760, India; 8Laboratoire des Substances Naturelles, Institut National de Recherche et d’Analyse Physico-Chimique, Biotechpôle de Sidi Thabet 2020, Tunisia; 9Department of Pharmacy, University of Salerno, Via Giovanni Paolo II, 132, Fisciano, 84084 Salerno, Italy; 10Faculty of Science and Arts in Baljurashi, Albaha University, P.O. Box 1988, Albaha 65527, Saudi Arabia; 11Faculty of Science of Sfax, Department of Chemistry, University of Sfax, B.P. 1171, Sfax 3000, Tunisia

**Keywords:** *Ducrosia flabellifolia*, essential oil, methanolic extract, antimicrobial, antioxidant, anticancer, ADME, *in-silico*, molecular interactions

## Abstract

In the present study, the chemical composition of the volatile oil and methanolic extract from *Ducrosia flabellifolia* Boiss. was investigated. The antimicrobial, antioxidant, and anticancer activities of the methanolic extract from *D. flabellifolia* aerial parts were screened using experimental and computational approaches. Results have reported the identification of decanal (28.31%) and dodecanal (16.93%) as major compounds in the essential oil obtained through hydrodistillation. Farnesyl pyrophosphate, Methyl 7-desoxypurpurogallin-7-carboxylate trimethyl ether, Dihydro-Obliquin, Gummiferol, 2-Phenylaminoadenosine, and 2,4,6,8,10-dodecapentaenal, on the other hand, were the dominant compounds in the methanolic extract. Moreover, the tested extract was active against a large collection of bacteria and yeast strains with diameter of growth inhibition ranging from 6.67 ± 0.57 mm to 17.00 ± 1.73 mm, with bacteriostatic and fungicidal activities against almost all tested microorganisms. In addition, *D. flabellifolia* methanolic extract was dominated by phenolic compounds (33.85 ± 1.63 mg of gallic acid equivalent per gram of extract) and was able to trap DPPH• and ABTS•+ radicals with IC_50_ about 0.05 ± 0 mg/mL and 0.105 ± 0 mg/mL, respectively. The highest percentages of anticancer activity were recorded at 500 µg/mL for all cancer cell lines with IC_50_ about 240. 56 µg/mL (A-549), 202.94 µg/mL (HCT-116), and 154.44 µg/mL (MCF-7). The *in-silico* approach showed that *D. flabellifolia* identified compounds bound 1HD2, 2XCT, 2QZW, and 3LN1 with high affinities, which together with molecular interactions and the bond network satisfactorily explain the experimental results using antimicrobial, antioxidant, and anticancer assays. The obtained results highlighted the ethnopharmacological properties of the rare desertic *D. flabellifolia* plant species growing wild in Hail region (Saudi Arabia).

## 1. Introduction

The *Ducrosia flabellifolia* Boiss. (*D. flabellifolia*) plant species belongs to Apiaceae family and is popularly known in Saudi Arabia as Haza [1]. The aerial parts of this species are smoked in form of cigarettes and have been used in traditional medicine by the local people as a sedative agent, and for the treatment of dental pain [2]. Volatile oil of *D. flabellifolia* has been reported for its potent antimicrobial potential towards *C. albicans* and *S. aureus* and moderate activity against *E. coli* and *P. aeruginosa*. The essential oil of *D. flabellifolia* has been documented for its moderate-to-weak anti-proliferative activity against three human cancer cell lines, MCF-7, K562 and LS180 [3]. In addition, the *D. flabellifolia* ethanol extract has been reported for its apoptotic effect toward breast cancer [4]. More recently, it has been demonstrated that D. *flabellifolia* hydroalcoholic extract collected from Hail region (Saudi Arabia) is a rich source of chlorogenic acid, ferulic acid, caffeic acid, and sinapic acid with good antimicrobial and antioxidant activities [5].

Cancer as a non-communicable disease was classified as the most common cause of death after cardiovascular diseases [6]. According to the data from the international agency for research on cancer (IARC), in Saudi Arabia, the incidence of different types of cancers has increased over the past decade and the number of new cancer cases was estimated to be 27,885, including 13,069 deaths with colorectum, breast, thyroid, non-Hodgkin and lymphoma remaining the most common type of cancers [6]. The drastic spread of cancer disease with high mortality rate is due to its ability to metastasize and its migrating effect among multiple organs [7]. Despite the current treatments, it is still considered the second most devastating cause of death worldwide [8,9]. On the other hand, the emergence of multidrug resistance (MDR) causing infections has promoted the development of novel antibacterial agents [10,11]. Infections and cancerous diseases are enhanced and aggravated by oxidative stress, which is a key factor due to its dramatic and fulgurant impact on the generation of free radicals which overcome suffering and stimulate various diseases [12]. Moreover, the increase in the number of infections caused by pathogenic microorganisms in cancer patients has encouraged scientists to search for novel therapeutic agents [13]. Conventional cancer chemotherapeutic drugs are based on the administration of drugs with the power to hinder the proliferation of tumor cells by inducing their apoptosis. They are mostly non-selective to cancer cells with detrimental side effects that cause serious health diseases, and multidrug-resistant microorganisms that contribute to the development of drug resistance in cancer cells [14,15].

For the above-mentioned reasons, and due to a lack of new anti-cancer and anti-infective agents with high therapeutic efficacy, no drug resistance and low side effects, the exploration of a bioassay-guided approach for the discovery of anti-cancer and antimicrobial natural products has been intensified and become a focus of interest for scientists and pharmaceutical research. Herbal treatment is the greatest gift that humans can use to improve their health [16,17,18]. Moreover, aromatic and medicinal herbs still remain the ultimate choice and a source of promising molecules in all areas of health, both in the treatment and prevention of certain pathologies [19]. 

Based on traditional claims regarding the use of *D. flabellifolia*, the purpose of the present study was to assess for the first time the phytochemical composition, the in vitro antioxidant, and antimicrobial activities of *D. flabellifolia* methanolic extract and its anti-proliferative effectiveness against colon, lung, and breast cancer cell lines, supported with the ADME and molecular docking studies of their major constituents. The latter focused on targeting 1HD2, 2XCT, 2QZW, AND 3LN1 macromolecules, which correspond to Human peroxiredoxin 5, *S. aureus* Gyrase complex, Secreted aspartic proteinase (Sap) 1 from *C. albicans*, and cyclooxygenase-2 (COX-2) to assess the antioxidant, antibacterial and anticancer/anti-inflammatory potentials, respectively.

## 2. Results

### 2.1. Chemical Composition of D. flabellifolia EO

The yield of extraction of the essential oil was about 0.24 ± 0 mL/100 g of plant material. In addition, out of 98.46% of bioactive compounds identified with GC-MS technique, miscellaneous compounds are the dominant group (64.10%), followed by monoterpene hydrocarbons (16.60%), oxygenated monoterpenes (10.89%), and oxygenated monoterpenes (6.87%). The main most volatile constituents were decanal (28.31%) and dodecanal (16.93%), and 1-heptadecne (8.30%) were the dominant compounds followed by β-eudesmol (6.87%), α-pinene (5.83%), β-phellandrene (5.76%), and 1-decanol (4.34%) (Table 1).

### 2.2. Chemical Composition of D. flabellifolia Methanolic Extract

The obtained methanolic extract was oily with a black color. The yield of extraction was about 20.06 ± 0.19 g of dry extract/100 g of plant material. Seventeen tripeptides with molecular weights ranging from (288.1552) to (432.2024) g/mol were tentatively identified through comparison of spectrum data of the extract with that of known compounds. Details of identified peptides are given in Appendix A.

It is important to note that all compounds were first reported in this study for *D. flabellifolia* aerial parts methanolic extract analyzed with HR-LC/MS. The complete list of identified chemical bioactive compounds is summarized in Table 2. 

### 2.3. Antimicrobial Activities of D. flabellifolia Methanolic Extract

The antimicrobial activities of *D. flabellifolia* methanolic extract was assessed using disc diffusion and microdilution assays. Results summarized in Table 3 showed high diameter of growth inhibition zones (mGIZ ± SD) ranging from 10.33 ± 0.57 mm (*P. aeruginosa*; Environmental strain, pf8) to 14.67 ± 0.57 mm (*S. aureus* MDR, Clinical strain, 136). The highest mGIZ was recorded for *C. neoformans* (17.00 ± 1.73 mm) and *C. albicans* ATCC 10231 (16.33 ± 0.57 mm). However, *C. vaginalis* (6.00 ± 0 mm), *Candida* sp. (6.67 ± 0.57 mm), *A. fumigatus* ATCC 204305 (8.33 ± 1.15 mm), and *A. niger* (8.67 ± 0.57 mm) were the most resistant microorganisms. Using the MBC/MIC ratio, the tested extract showed bacteriostatic action against almost all tested bacterial strains (MBC/MIC ratio > 4) with the exception against *S. aureus* MDR (Clinical strain, 136), *S. paucimobilis* (Clinical strain, 144), and *A. baumannii* (Clinical strain, 146) with MBC/MIC ration lower than 4 highlighting a bactericidal action against these bacteria. Interestingly, *D. flabellifolia* methanolic extract exhibited fungicidal activity against the four yeast strains tested (MFC/MIC < 4).

### 2.4. Antioxidant Activities of D. flabellifolia Methanolic Extract

Total phenolic content (TPC), total flavonoids content (TFC), and total tannins content (TTC) were estimated, and the obtained results revealed a dominance of phenolic compounds (TPC = 38.85 ± 1.63 mg of gallic acid equivalent per gram of extract) followed by flavonoids (TFC = 17.06 ± 0.48 mg of quercetin equivalent per gram of dry extract), and condensed tannins (TTC = 7.80 ± 0.69 mg of tannic acid equivalent per gram of dry extract). In addition, *D. methanolic* extract was able to trap DPPH^•^ and ABTS^•+^ radicals with IC_50_ about 0.05 ± 0 mg/mL and 0.105 ± 0 mg/mL, respectively (Table 4).

Using the Duncan test, a significant difference (*p* < 0.005) was found between *D. flabellifolia* methanolic extract and the standards molecules used (BHT and AA) in all three antioxidants systems used. Using β-carotene/linoleic assay, our results indicated that high concentration from *D. flabellifolia* methanolic extract was needed for bleaching 50% of (IC_50_ = 5.00 ± 0.78 mg/mL) as compared to BHT (IC_50_ = 0.042 ± 0 mg/mL) and AA (IC_50_ = 0.017 ± 0 mg/mL).

### 2.5. Anticancer Activities of D. flabellifolia Methanolic Extract

The anticancer activity of *D. flabellifolia* methanolic extract was tested against breast (MCF-7), lung (A549), and colon (HCT-116) cancer cell lines using the MTT assay (Figure 1). Results showed an increase in cell viability inhibition in a concentration dependent manner. The highest percentages were recorded at 500 µg/mL for all cancer cell lines with IC_50_ about 240.56 µg/mL (A-549), 202.94 µg/mL (HCT-116), and 154.44 µg/mL (MCF-7).

### 2.6. ADME Predictions

In drug development, it is imperative to study and investigate its safety and efficacy to know how a successful drug processes and reacts with the human body. For this, the major compounds were predicted for their absorption, distribution, metabolism, and excretion (ADME) properties tool to verify that the designed molecules are viable drugs. The selected phytocompounds showed good bioavailability score, high gastrointestinal absorption (GI) and some of them were predicted to be blood-brain-barrier (BBB) permeant. Except compound 9 (Ergoline-1,8-dimethanol, 10-methoxy-6-methyl-, (8b)-), the rest have been predicted to be not P-gp substrate, meaning that they are likely to have promising intestinal absorption and bioavailability. Simultaneously, predictive data showed that most of the selected compounds are not inhibitors of cytochrome P450 isoenzymes CYP 1A2, CYP2C19, CYP2C9, CYP2D6 and CYP3A4, meaning that they will not hamper the biotransformation of drugs metabolized by CYP450 enzymes. From the skin permeation (LogKP), we deduced acceptable values. All examined compounds were predicted to comply with Lipinski’s rule-of-five, suggesting their good drug-likeness behavior. They also do not violate Ghose, Veber, Egan or Muegge (with some irregularities) filters. 

The bioavailability radar of the selected analogues showed that the colored zone is the desired physicochemical space for good oral bioavailability in which the following properties were taken into account: flexibility, lipophilicity, saturation, size, polarity and solubility (Table 5). As shown in Figure 2, most of them fall entirely within the pink area, suggesting that they are suitable for better bioavailability. 

### 2.7. Molecular Docking Study

Table 6 and Appendix A showed that *D. flabellifolia* compounds bound the four targeted receptors with negative free binding energy but with different scores, except for Khayanthone for 1HD2 macromolecule. While the best binding score was predicted in the 3LN1-Harderoporphyrin complex, it ranged between –3.7 and –9.9 kcal/mol for the other complexes. Harderoporphyrin, which possessed the best binding score with 3LN1, was also predicted to interact with 1HD2, 2XCT and 2QZW receptors with acceptable binding scores of –6.4, –8.2 and –8.2 kcal/mol and included several different key residues. It was also deeply embedded into these targeted receptors and showed 2.047, 2.186, 1.804 and 1.898 Å only, respectively, for 1HD2, 2XCT, 2QZW, and 3LN1.

*D. flabellifolia* compounds were also found to be deeply embedded in all the studied receptors (1HD2, 2XCT, 2QZW, AND 3LN1). In this context, 1.476 Å only was reported in the compound (Khayanthone) while docked to the 1HD2 receptor. The bond network included H-bonds, which are commonly evaluated to assess the biological activities of the assessed compounds, associated several hydrophobic bonds: Pi-anion, Pi-cation, Pi-alkyl, and Pi-Pi T-shaped, as shown by the corresponding diagram of interactions (Figure 3).

## 3. Discussion

The yield of volatile oil was about 0.24 ± 0 mL/100 g of plant material, while the yield of methanolic extract was about 20.06 ± 0.19 g of dry extract/100 g of plant material. In this study, we report the identification of twenty-two bioactive compounds in the essential oil of *D. flabellifolia* aerial parts obtained by hydrodistillation. In fact, *D. flabellifolia* was dominated by decanal, dodecanal, 1-heptadecene, and α-pinene. Linear furanocoumarins have been previously identified as the major non-volatile components of the *Ducrosia* genus [20], whilst the main most volatile constituents were aliphatic hydrocarbons, with decanal (10.1–74.0%) and dodecanal (7.2–33.41%) as the major constituents [1]. Our results are in accordance with those reported by Al-Ghamdi et al. [21], who described the identification of 52 compounds in stems, leaves, and flowers of *D. flabellifolia* collected from northern border (Saudi Arabia). The same authors reported that the obtained essential oil was dominated by aldehyde hydrocarbons (stems 65.11%, leaves 65.39%, and flowers 67.30%), and decanal and dodecanal were the main identified compounds in the three tested organs. Similarly, Shahabipour and colleagues [3] reported that n-decanal (32.80%), dodecanal (32.6%), n-decanol (4.30%) and (2E)-tridecen-1-al (3.30%) were the dominant compounds identified in *D. flabellifolia* essential oil from Iran. The essential oils (fresh leaves and flowers) from Jordanian *D. flabellifolia* plant species obtained through hydrodistillation and solid phase microextraction (SPME) was a rich source of aliphatic compounds where n-decanal was the predominant compound obtained in *D. flabellifolia* fresh leaves (Hydrodistillation: 36.61%; SPME: 24.44%), while n-decanol was the dominant bioactive compound in the essential oils of dry leaves (27.88%) and fresh flowers’ (11.49%) essential oils extracted by SPME technique [2]. 

In addition, as regards *D. flabellifolia* methanolic extract, we reported in this study the tentative identification of seventeen small peptides (tripeptides) and twenty-two phytochemical compounds (mainly farnesyl pyrophosphate, Methyl 7-desoxypurpurogallin-7-carboxylate trimethyl ether, Dihydro-Obliquin, Gummiferol, 2-Phenylaminoadenosine, and 2,4,6,8,10-dodecapentaenal) using HR-LCMS techniques. In fact, our team reported the identification of twenty-three bioactive compounds in *D. flabellifolia* hydroalcoholic extract dominated by (mg/Kg of crude extract ) chlorogenic acid (5980.96 ± 73.12), ferrulic acid (180.58 ± 2.77), caffeic acid (70.90 ± 1.75), sinapic acid (61.74 ± 2.79), 2-5 dihydrobenzoic acid (59.74 ± 0.945), *p*-coumaric acid (55.11 ± 0.765), and 2-Hydroxycinnamic acid (31.28 ± 0.015) analyzed using liquid chromatography-electrospray tandem mass spectrometry [5]. Previous works reported the isolation of tetradecenol from leaves and fruit methanolic extract of *D. anethifolia* collected from the province Thadeq (180 km north of Riyadh) by using GC-MS technique [22]. Moreover, *D. anethifolia* is considered a good source of biologically active compounds especially coumarins and furanocoumarin [23]. 

We report also in this study that using Parveen scheme [24], *D. flabellifolia* methanolic extract was moderately to highly active against several Gram-positive and Gram-negative bacteria, yeast, and mold strains. In fact, on agar plates, the mean diameter of growth inhibition zone ranged from 10.33 ± 0.57 mm for *P. aeruginosa* to 14.67 ± 0.57 mm for *S. aureus* MDR strains. In addition, *C. albicans* ATCC 10231 and *C. neoformans* ATCC 14116 were the most sensitive yeast strains with mGIZ about 16.33 ± 0.57 mm and 17.00 ± 1.73 mm, respectively. Moreover, using the scheme proposed by Gatsing et al. [23] and Moroh et al. [25], *D. flabellifolia* methanolic extract exhibited bacteriostatic activity against almost all tested bacteria with the exception of *S. aureus*, *S. paucimobilis*, and *A. baumannii* (MBC/MIC = 4). Interestingly, *D. flabellifolia* exhibited fungicidal character against the four tested yeast strains (MFC/MIC ≤ 4). More recently, we reported that hydroalcoholic extract from *D. flabellifolia* collected from the Hail region (Saudi Arabia) was active against ESKAPE pathogens in a concentration dependent manner [5]. In addition, high concentrations (from 100 to 200 mg/mL) of *D. flabellifolia* methanol–water extract were needed to kill *C. utilis* ATCC 9255, *C. tropicalis* ATCC 1362, *C. guillermondii* ATCC 6260, and *C. albicans* ATCC 20402 [5]. 

Previous reports have demonstrated that extracts from *Ducrosia* plant species possessed good-to-moderate susceptibility against *S. epidermidis* ATCC 49461, *B. cereus* ATCC 10876, *S. aureus* clinical isolate and *S. aureus* ATCC 25923 [26]. Similarly, Alsaggaf [27] reported that *D. anethifolia* extract was able to inhibit the growth of both methicillin sensitive and methicillin resistant *S. aureus* strains with a diameter of growth inhibition zone ranging from (7.8 ± 0.4) mm for MRSA2 strain to (9.6 ± 0.6) mm, as compared to *S. aureus* ATCC 25923 (10.2 ± 0.5 mm). Al-Whibi and colleagues [22] studied the antimicrobial activities of *D. anethifolia* (leaves and fruit methanolic and acetone extracts) against *S. aureus* ATCC 25923, methicillin resistant *S. aureus* MRSA ATCC 12498, *B. subtilis* (ATCC 6633), *E. feacalis* ATCC 29122, *E. coli* ATCC 25966, *P. aeruginosa* ATCC 27853, *K. pneumoniae* ATCC 700603, *Salmonella* sp. and *Serratia* sp.

We also tested the antioxidant activities of *D. flabellifolia* methanolic extract using DPPH, ABTS, and β-carotene assays. Results obtained showed good ability to scavenge the three radicals at low IC_50_ values. Similar results were obtained with *D. flabellifolia* methanol/water extract (Table 7). In fact, low concentration from methanol/water extract was able to scavenge DPPH^•^ free radicals (IC_50_= 0.014 ± 0.045 mg/mL) as compared to methanolic extract (IC_50_ = 0.048 ± 0.004 mg/mL).

Our results also indicated that *D. flabellifolia* methanolic extract exhibited anticancer activities against colon, lung, and breast cancer cell lines with IC_50_ about 240.56 µg/mL (A-549), 202.94 µg/mL (HCT-116), and 154.44 µg/mL (MCF-7). Previous reports have demonstrated that essential oil from *Ducrosia* members (*D. flabellifolia* and *D. anethifolia*) exhibited cytotoxic activity against human chronic myelogenous leukemia cell lines (K562), human colon adenocarcinoma (LS180), and human breast adenocarcinoma (MCF-7) [22]. The same authors reported that *D. flabellifolia* essential oil was active against K562, LS180, and MCF-7 cell lines with IC_50_ value about 304.0 ± 87.2 µg/mL, 286.9 ± 28.0 µg/mL, and 511.2 ± 133.2 µg/mL, respectively [22]. It has been also demonstrated *that D. ismaelis* essential oil (Decanal 40.6%, α-pinene 15.1%, and dodecanal 13.7%) exhibited anticancer activities against three cell lines, namely MCF-7 (IC_50_ 66.24 ± 1.26 µg/mL), LoVo (IC_50_ 102.53 ± 1.00 µg/mL), and HepG2 (IC_50_ 137.32 ± 2.48 µg/mL) [28].

The ADME prediction revealed that all the studied compounds possessed high GI absorption associated variable BBB permeation. Most of the selected compounds were also found to not inhibit the cytochrome P450 isoforms (1A2, 2C19, 2C9, 2D6 and 3A4), which indicates the safe use of these compounds and the absence of any disruption in drug distribution and metabolism [29,30,31]. It was noticed that bioavailability scores ranged between 0.55 and 0.85 for the major *D. flabellifolia* studied compounds, which indicate acceptable bioavailability associated potential physiological activity of these phytochemicals as previously reported by several studies on natural and synthesized compounds [29,32,33]. The acceptable bioavailability scores were confirmed by the polygons illustrations. In fact, most of the compounds physicochemical properties stayed in the pick areas that indicate the most suitable oral bioavailability. 

The *D. flabellifolia* identified compounds were subjected to computational assay to assess their molecular interactions with some key receptors related to antimicrobial, antioxidant, and anticancer activities. Molecular docking results showed that all compounds bound the four targeted receptors with negative free binding energy but with different scores, except Khayanthone for 1HD2 macromolecule. The free binding energy ranged between –3.7 and –9.9 kcal/mol. It has been reported that variation in such score values is mainly linked to the 3D chemical structures of the ligands [29,30,31,32,33] The best binding score was predicted in 3LN1-Harderoporphyrin complex. The same compound (Harderoporphyrin) was also found to interact with each of 2XCT and 2QZW receptors with an interesting bound energy of –8.2 kcal/mol. The molecular interactions of *D. flabellifolia* compounds with the targeted receptors included up to twelve conventional H-bonds and involved several different key residues. In addition, *D. flabellifolia* compounds were also found to be deeply embedded in all the studied receptors. The lowest distance of 1.476 Å only was reported in the compound (Khayanthone) while docked to the 1HD2 macromolecule. Regardless of H-bonds that are commonly evaluated to assess the biological activities of the targeted compounds, a network of hydrophobic bonds was also found within the different studied complexes. This may contribute to the stability of the complexes as reported in several recent *in silico* studies [34,35,36]. Our results exhibit that all *D. flabellifolia* identified compounds established acceptable number of H-bonds. The corresponding diagram of interactions of the selected established complexes (Figure 4) showed involvement of several key residues and diversified bond network: Pi-anion, Pi-cation, Pi-alkyl, Pi-Pi T-shaped…, which support the H-bonds and contributed to the complex stability [32,33]. Interactions with key residues was found to promote biological activities including antimicrobial, antioxidant, and anticancer potential of the studied compounds. In this context, all *D. flabellifolia* compounds were found to be in close proximity of all the targeted receptor with distance less than 3 Å. Ligands deeply embedding were reported to enhance the biological activity [30,31]. 

Overall, the high antimicrobial, antioxidant, and anticancer activities of the tested *D. flabellifolia* methanolic extract can be attributed to its richness in bioactive compounds belonging to different chemical classes, such as alkaloids, coumarins, polyphenols, fatty acyls, and terpenoids. Docking results revealed that the high molecular interactions obtained justify that the antimicrobial, antioxidant, and anticancer potentials of the studied *D. flabellifolia* are thermodynamically possible and this could explain the results obtained in vitro.

## 4. Materials and Methods

### 4.1. Plant Material Sampling

In this study, *D. flabellifolia* Boiss. locally known as Al-Hazaa (Figure 4), was collected from Al-Mu’ayqilat, 27°16′41.9″ N, 41°22′48.0″ E in October 2019. The plant material was air-dried at room temperature for one week. The methanolic extract was obtained using maceration technique (20 g of powdered aerial parts in 200 mL of pure methanol at room temperature for 72 h with low agitation). The filtrate was recuperated through lyophilization and kept at −4 °C until use.

### 4.2. Phytochemical Composition 

#### 4.2.1. Composition of the Essential Oil 

The gas chromatography–mass spectrometry (GC–MS) analyses were performed on a gas chromatograph HP 6890 (II) interfaced with an HP 5973 mass spectrometer (Agilent Technologies, Palo Alto, CA, USA) with electron impact ionization (70 eV). The volatile compounds were identified by comparing their retention indices relative to (C7–C20) n-alkanes with those of literature and/or with those of authentic compounds available in our laboratory, and by matching their mass spectral fragmentation patterns with corresponding data (Wiley 275.L library) and other published mass spectra [37], as well as by comparison of their retention indices with data from the Mass Spectral Library.

#### 4.2.2. Composition of the Methanolic Extract

The identification of phytoconstituents in the methanolic extract from *D. flabellifolia* methanolic extract was performed using High Resolution-Liquid Chromatography Mass Spectroscopy (HR-LCMS) as previously described by Noumi et al. [38]. MS data were provided in negative and positive ionization mode.

### 4.3. Antimicrobial Activities of D. flabellifolia Methanolic Extract 

*D. flabellifolia* methanolic extract was tested for its ability to inhibit the growth of twelve clinical and environmental bacterial strains (*Escherichia coli* ATCC 35218, *Pseudomonas aeruginosa* ATCC 27853, *Proteus mirabilis* ATCC 29245, *Klebsiella pneumoniae* ATCC 27736, *P. mirabilis*, *Staphylococcus sciuri*, *Streptococcus pyogens*, *P. aeruginosa*, *S. aureus* MDR, *Enterobacter cloacae*, *Stenotrophomonas paucimobilis*, *Acinetobacter baumannii*), four yeasts (*Candida albicans* ATCC 10231, *Cryptococcus neoformans* ATCC 14116, *C. vaginalis*, *Candida* sp.), and two mold strains (*Aspergillus fumigatus* ATCC 204305 and *A. niger*). Disc diffusion assay was used (20 µL/disc) using the same protocol described by Snoussi et al. [5] for the determination of the diameter of growth inhibition zone estimated on agar medium (Mueller Hinton for bacteria and Sabouraud Chloramphenicol agar for fungi). Parveen et al. [24] was used to interpret the obtained. Ampicillin and Amphotericin B were used as control. 

To estimate the minimal inhibitory concentrations (MICs values expressed in mg/mL) and minimal bactericidal/fungicidal concentration (MBCs and MFCs values), the obtained extract was serially diluted in DMSO-5% supplemented with Tween 80 (From 100 mg/mL to 0.097 mg/mL) in 96-well microtiter plates containing 95 µL of the microbial suspension and 95 µL of the enrichment broth (Lauria Bertani for bacteria and Sabouraud dextrose broth for fungal strains). To interpret the character of the tested extract, we used the ratios (MBC/MIC ratio and MFC/MIC ratio) described by Gatsing et al. [23] and Moroh et al. [17].

### 4.4. Antioxidant Activities 

The ability of Al-Haza extract against DPPH-H was determined following the same method as Mseddi al. [39], and the method of Koleva et al. [40] for β-Carotene bleaching test. The radical scavenging activity against ABTS•+ (2,2′-azino-bis(3-ethylbenzothiazoline-6-sulfonic acid)) radical cations was measured using the same protocol described by Hamdi et al. [41].

### 4.5. Anticancer Activity

*D. flabellifolia* methanolic extract was tested against human lung (A549), breast (MCF-7), and colon (HCT-116) cancer cells at different concentrations. The percentage growth inhibition was calculated after subtracting the background and the blank, and the concentration of the test drug needed to inhibit cell growth by 50% (IC_50_) was calculated from the dose–response curve for the respective cell line [42].

### 4.6. Computational Study

#### 4.6.1. ADME Properties

The pharmacokinetic properties, drug-likeness, and medicinal properties of the identified bioactive molecules from *D. flabellifolia* methanolic extract were studied using the same recommendations described by Daina and colleagues [43]. Each structure was imported, and the structure SMILES was entered at the interface of the website (http://swissadme.ch/, accessed on 2 October 2022), a free web tool to assess the pharmacokinetics, drug-likeness and medicinal chemistry friendliness of small molecules. The SwissADME drug design study was run and the ADMET properties/parameters were generated.

#### 4.6.2. Molecular Docking Study

Four different receptors, (PDB ID) 2XCT (*S. aureus* IIA topoisomerase), 2QZW (*C. albicans* Sap1), 1HD2 (human peroxiredoxin 5 protein) and 3LN1 (cyclooxygenase-2; COX-2), have been targeted to check the potential antimicrobial, antioxidant, and anticancer effect of the *D. flabellifolia* identified compounds. The crystalized structures of the selected receptors have been obtained from RCSB protein data bank. ChemDraw was used to obtain the chemical structures whenever needed, following the pre-processing of both ligands and receptors (removal of water molecules and addition of polar hydrogens and Koleman charges) using Autodock vina packages v1.2.3 [30,31]. The binding scores and calculation of embedding distances and bonding network were studied as previously described based on the CHARMM force field [32,33,34,35]. The reason behind the selection of these receptors is their involvement in antioxidant and anticancer pathways and the fact that they are commonly targeted in pharmaceutical and drug design approaches [33].

### 4.7. Statistical Analysis

Experiments were performed in triplicate and average values were calculated using the SPSS 25.0 statistical package for Windows. Duncan’s multiple-range tests for means with a 95% confidence interval (*p* ≤ 0.05) was used to calculate the differences in means.

## 5. Conclusions

In conclusion, our results indicated that fresh aerial parts of *D. flabellifolia* growing wild in the Hail region possess antimicrobial activity against several Gram-positive and Gram-negative bacteria, yeast, and molds with different degree. In fact, their phytochemical composition revealed the presence of various compounds with known biological properties in both essential oil and methanolic extract. The pharmacokinetic and ADMET properties of *D. flabellifolia* phytochemicals may explain the in vitro antimicrobial, antioxidant, and anticancer findings, which may result from the potential molecular interactions of these chemicals with the concerned receptors (1HD2, 2XCT, 2QZW, and 3LN1). These results support the benefits of this medicinal plant as a source of bioactive molecules for different ethnobotanical uses.

## Figures and Tables

**Figure 1 metabolites-13-00064-f001:**
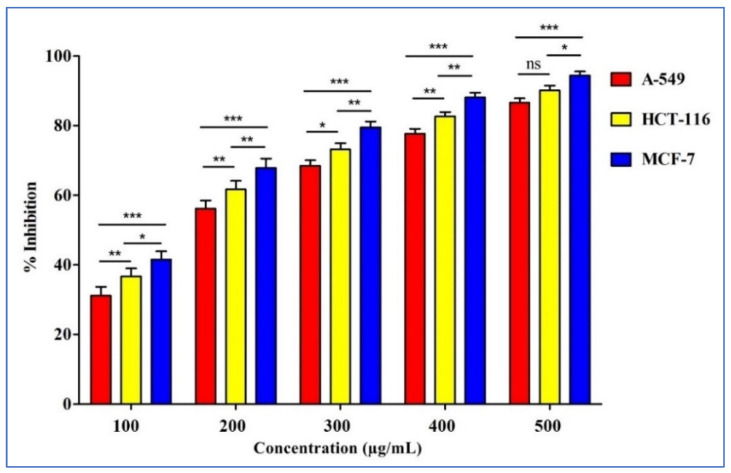
Effect of different concentrations from *D. flabellifolia* methanolic extract on breast (MCF-7), lung (A549), and colon (HCT-116) cancer cell lines. Error bars indicate SDs (± standard deviation) of three independent experiments. Significance; ns > 0.01, * *p* < 0.01, ** *p* < 0.001, *** *p* < 0.0001.

**Figure 2 metabolites-13-00064-f002:**
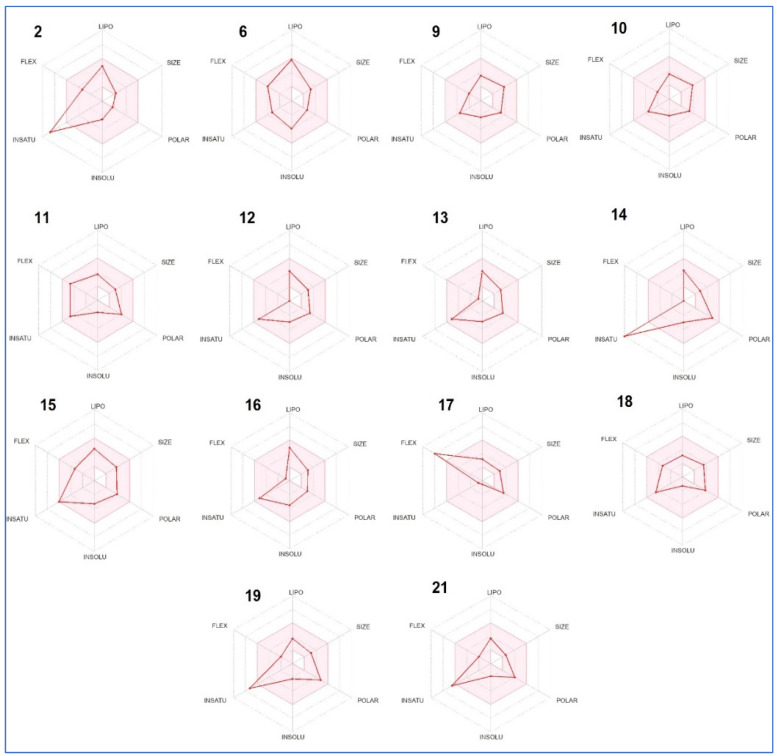
Radar plot of the selected peptides and phytoconstituents based on physicochemical indices ideal for oral bioavailability.

**Figure 3 metabolites-13-00064-f003:**
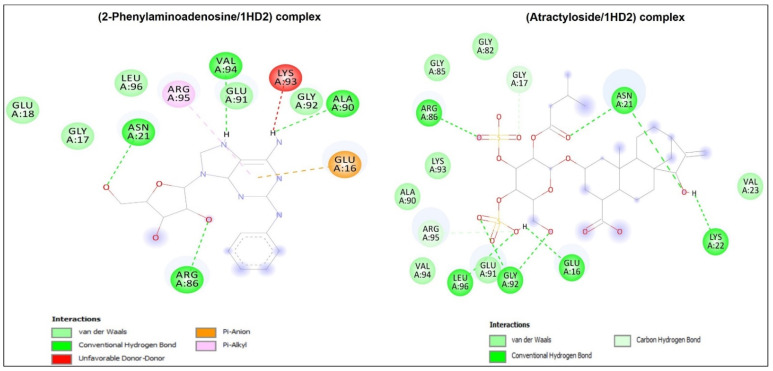
Three-dimensional (3D) residual interactions network of the best selective compounds with the active site of human peroxiredoxin 5 protein (PDB ID: 1HD2), *C. albicans* Sap1 (PDB ID: 2QZW), and *S. aureus* IIA topoisomerase (PDB ID: 2XCT).

**Figure 4 metabolites-13-00064-f004:**
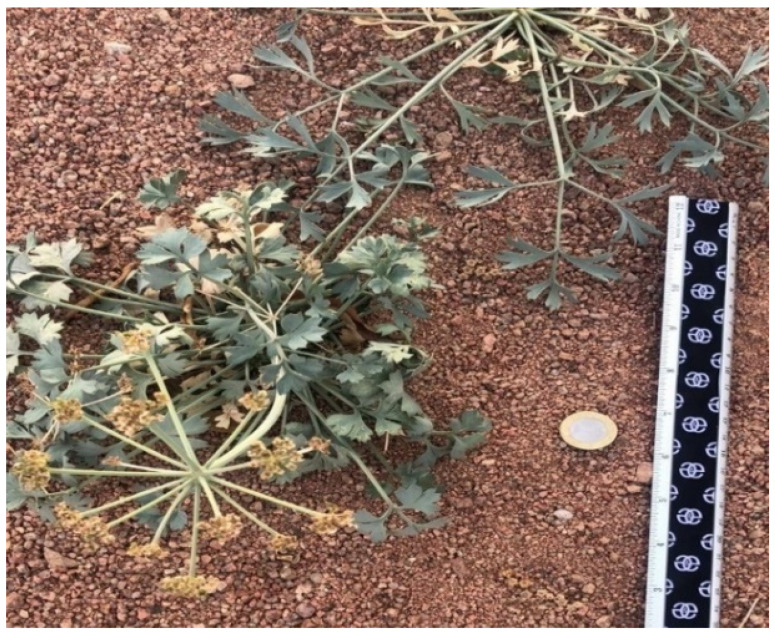
Al-Haza plant species collected from Al-Mu’ayqilat (Hail region).

**Table 1 metabolites-13-00064-t001:** Chemical composition of *D. flabellifolia* aerial parts identified by GC-MS technique.

N°	Compound	RI *	*D. flabellifolia* EO	Molecular Weight	Chemical Formula
1	Nonane	900	0.68	128.259	C_9_H_20_
2	α-pinene	940	5.83	136.238	C_10_H_16_
3	Sabinene	969	0.85	136.23	C_10_H_16_
4	o-cymene	976	0.51	134.22	C_10_H_14_
5	β-pinene	983	0.38	136.278	C_10_H_16_
6	β-myrcene	992	2.87	136.238	C_10_H_16_
7	α-phellandrene	1005	0.40	136.23	C_10_H_16_
8	β-phellandrene	1029	5.76	136.23	C_10_H_16_
9	Fenchone	1093	1.81	152.23	C_10_H_16_O
10	4-undecene	1076	0.72	154.292	C_11_H_22_
11	Undecane	1100	0.33	156.313	C_11_H_24_
12	Citronellal	1152	1.81	154.25	C_10_H_18_O
13	Decanal	1202	28.31	153.26	C_10_H_20_O
14	Citronellol	1236	1.95	156.269	C_10_H_20_O
15	Verbenyl acetate	1269	2.84	194.270	C_12_H_18_O_2_
16	1-decanol	1274	4.34	158.28	C_10_H_21_OH
17	Thymol	1298	1.84	150.22	C_10_H_14_O
18	Undecanal	1307	0.26	170.296	C_11_H_22_O
19	Geranyl acetate	1382	0.64	196.29	C_10_H_20_O_2_
20	Dodecanal	1412	16.93	184.32	C_12_H_24_O
21	2-dodecenal	1476	0.80	182.3	C_12_H_22_O
22	1-hexadecene	1593	1.59	224.42	C_16_H_32_
23	Tetradecanal	1614	1.14	212.37	C_14_H_28_O
24	β-eudesmol	1654	6.87	222.37	C_15_H_26_O
25	1-heptadecne	1697	8.30	238.5	C_17_H_34_
26	2-Hydroxycyclopentadecanone	1852	0.70	240.38	C_15_H_28_O_2_

* RI: Retention index relative to (C_8_-C_24_) *n*-alkanes on HP-5MS column.

**Table 2 metabolites-13-00064-t002:** Phytochemical compounds identified by the HR-LCMS technique in *D. flabellifolia* methanolic extract.

N°	Compound Name	Chemical Class	RT (mn)	MW (g/mol)	Chemical Formula	[*m/z*]-	[*m/z*]+
**1**	10-Hydroxyloganin	Terpenoids	0.963	406.1437	C_17_ H_26_ O_11_	387.1277	-
**2**	2,4,6,8,10-dodecapentaenal	Fatty Acyls	1.060	174.105	C_12_ H_14_ O	191.0638	-
**3**	2-Phenylaminoadenosine	Glycosides	3.827	358.1398	C_16_ H_18_ N_6_ O_4_	357.1324	-
**4**	Atractyloside	Glycosides	4.260	726.2192	C_30_ H_46_ O_16_ S_2_	707.201	-
**5**	Cortisol 21-sulfate	Sterols lipids	5.549	442.1627	C_21_ H_30_ O_8_ S	459.1217	-
**6**	5,8,11-heptadecatriynoic acid	Fatty Acyls	6.468	258.156	C_17_ H_22_ O_2_	239.1382	-
**7**	Galactan	Polysaccharides	8.436	680.2045	C_24_ H_40_ O_22_	679.199	-
**8**	Harderoporphyrin	Pigment	9.807	608.2509	C_35_ H_36_ N_4_ O_6_	643.2205	-
**9**	Ergoline-1,8-dimethanol, 10- methoxy-6-methyl-, (8b)-	Alkaloid	26.965	316.1812	C_18_ H_24_ N_2_ O_3_	297.1636	-
**10**	Ecgonine-methyl ester	Alkaloid	1.454	199.1195	C_10_ H_17_ N O_3_	-	200.1268
**11**	2-Hydroxy-3-(4-methoxyethylphenoxy)- propanoic acid	Organic Acids	5.585	240.101	C_12_ H_16_ O_5_	-	263.0902
**12**	Lomatin	Coumarins	6.222	246.878	C_14_ H_14_ O_4_	-	247.0951
**13**	Marmesin	Coumarins	7.088	246.0903	C_14_ H_14_ O_4_	-	269.0794
**14**	Purpurogallin	Natural Phenol	7.210	220.0361	C_11_ H_8_ O_5_	-	269.0794
**15**	Atranorin	Polyphenol	7.363	196.0387	C_9_ H_8_ O_5_	-	203.0328
**16**	Methyl 7- desoxypurpurogallin-7- carboxylate trimethyl ether	Natural Phenols	7.777	304.0935	C_16_ H_16_ O_6_	-	287.0903
**17**	Dihydro-Obliquin	Coumarin	8.789	246.0904	C_14_ H_14_ O_4_	-	269.0796
**18**	13-amino-tridecanoic acid	Fatty Acid	9.341	229.2031	C_13_ H_27_ N O_2_	-	230.2104
**19**	Gummiferol	Fatty Acyl	9.417	286.0836	C_16_ H_14_ O_5_	-	269.0803
**20**	Farnesyl pyrophosphate	Isoprenoid	9.786	382.128	C_15_ H_28_ O_7_ P_2_	-	405.1171
**21**	Syringic acid	Natural Phenols	15.399	198.054	C_9_ H_10_ O_5_	-	203.0326
**22**	Khayanthone	Polyphenols	18.482	570.2856	C_32_ H_42_ O_9_	-	593.275

**Table 3 metabolites-13-00064-t003:** Growth inhibition zone, MICs, MBCs, and MFCs values obtained using disc diffusion and microdilution assays.

**Code**	**Bacterial Strain**	***D. flabellifolia* Methanolic Extract**				**Ampicillin** **Mean ± SD (mm)**
		**mGIZ ± SD (mm)**	**MIC ^a^**	**MBC ^b^**	**MBC/MIC** **Ratio**	
B_1_	*E. coli* ATCC 35218	12.66 ± 1.15 ^bc^	12.50	200	16; bacteriostatic	7.00 ± 0.00 ^d^
B_2_	*P. aeruginosa* ATCC 27853	11.33 ± 0.57 ^cde^	25	200	8; bacteriostatic	7.33 ± 0.57 ^d^
B_3_	*P. mirabilis* ATCC 29245	12.67 ± 0.57 ^bc^	25	200	8; bacteriostatic	6.33 ± 0.57 ^d^
B_4_	*K. pneumoniae* ATCC 27736	14.33 ± 0.57 ^a^	25	200	8; bacteriostatic	6.66 ± 0.57 ^d^
B5	*P. mirabilis* (Environmental strain, 3)	12.67 ± 0.57 ^bc^	25	200	8; bacteriostatic	21.00 ± 1.00 ^a^
B6	*S. sciuri* (Environmental strain, 4)	11.33 ± 1.52 ^cde^	25	200	8; bacteriostatic	7.00 ± 0.00 ^d^
B7	*S. pyogens* (Clinical strain)	11.33 ± 1.15 ^cde^	25	200	8; bacteriostatic	16.00 ± 1.73 ^b^
B8	*P. aeruginosa* (Environmental strain, pf8)	10.33 ± 0.57 ^e^	12.50	100	8; bacteriostatic	6.66 ± 0.57 ^d^
B9	*S. aureus* MDR (Clinical strain, 136)	14.67 ± 0.57 ^a^	12.50	50	4; bactericidal	7.33 ± 0.57 ^d^
B10	*E. cloacae* (Clinical strain, 115)	14.33 ± 0.57 ^a^	12.50	100	8; bacteriostatic	6.66 ± 0.57 ^d^
B11	*S. paucimobilis* (Clinical strain, 144)	12.33 ± 0.57 ^cd^	25	100	4; bactericidal	7.66 ± 0.57 ^d^
B12	*A. baumannii* (Clinical strain, 146)	14.00 ± 0.00 ^ab^	12.50	50	4; bactericidal	13.33 ± 0.57 ^c^
**Code**	**Yeasts and molds**	**mGIZ±SD (mm)**	**MIC ^a^**	**MFC ^b^**	**MFC/MIC** **Ratio**	**Amphotericin B** **Mean ± SD (mm)**
Y_1_	*C. albicans* ATCC 10231	16.33 ± 0.57 ^a^	25	50	2; fungicidal	22.66 ± 1.15 ^a^
Y_2_	*C. neoformans* ATCC 14116	17.00 ± 1.73 ^a^	6.25	12.50	2; fungicidal	15.33 ± 0.57 ^b^
Y_3_	*C. vaginalis* (Clinical strain)	6.00 ± 0.00 ^d^	6.25	25	4; fungicidal	6.66 ± 0.57 ^d^
Y_4_	*Candida sp.* (Clinical strain)	6.67 ± 0.57 ^cd^	25	100	4; fungicidal	12.33 ± 0.57 ^c^
M_1_	*A. fumigatus* ATCC 204305	8.33 ± 1.15 ^bc^	-	-	-	15.00 ± 1.00 ^b^
M_2_	*A. niger*	8.67 ± 0.57 ^b^	-	-	-	6.00 ± 0.00 ^d^

Inhibition zone around the discs impregnated with *D. flabellifolia* methanolic extract (3 mg/disk) expressed as mean of three replicates (mm ± SD). SD: standard deviation. MIC: Minimal Inhibitory Concentration. MBC: Minimal Bactericidal Concentration. The letters (**a**–**e**) indicate a significant difference between the inhibition zones of the sample and amphotericin B against bacteria according to the Duncan test (*p* < 0.05).

**Table 4 metabolites-13-00064-t004:** Antioxidant activities of *D. flabellifolia* methanolic extract as compared to standard molecules.

Tests	DPPH IC_50_ (mg/mL)	ABTS IC_50_ (mg/mL)	β-Carotene IC_50_ (mg/mL)
*D. flabellifolia* methanolic extract	0.05 ± 0 ^a^	0.105 ± 0 ^a^	5.00 ± 0.78 ^a^
BHT (Butylated hydroxytoluene)	0.023 ± 0 ^b^	0.018 ± 0 ^b^	0.042 ± 0 ^b^
Ascorbic Acid	0.022 ± 0 ^b^	0.021 ± 0 ^b^	0.017 ± 0 ^b^

Letters (a,b) indicate a significant difference (*p* < 0.005) between *D. flabellifolia* methanolic extract and standard molecules.

**Table 5 metabolites-13-00064-t005:** Pharmacokinetics, drug-likeness and medicinal chemistry of identified compounds according to SwissADME software.

Entry	Bioactive Compounds													
	2	6	9	10	11	12	13	14	15	16	17	18	19	21
**Pharmacokinetics properties**														
GI absorption	High	High	High	High	High	High	High	High	High	High	High	High	High	High
BBB permeant	Yes	Yes	No	No	No	Yes	Yes	No	No	Yes	Yes	Yes	No	No
P-gp substrate	No	No	Yes	No	No	No	No	No	No	No	No	No	No	No
CYP1A2 inhibitor	No	Yes	No	No	No	Yes	Yes	No	No	Yes	Yes	No	No	No
CYP2C19 inhibitor	No	Yes	No	No	No	No	No	No	No	Yes	Yes	No	No	No
CYP2C9 inhibitor	No	Yes	No	No	No	No	No	No	No	Yes	No	No	No	No
CYP2D6 inhibitor	No	No	Yes	No	No	No	No	No	No	No	No	No	No	No
CYP3A4 inhibitor	No	No	No	No	No	No	No	Yes	No	No	No	No	No	No
Log Kp (cm/s)	−5.15	−4.60	−7.73	−7.08	−7.09	−6.45	−6.45	−6.18	−5.51	−6.44	−5.66	−7.66	−8.02	−6.77
**Druglikeness properties**														
Lipinski	Yes	Yes	Yes	Yes	Yes	Yes	Yes	Yes	Yes	Yes	Yes	Yes	Yes	Yes
Ghose	Yes	Yes	Yes	Yes	Yes	Yes	Yes	Yes	Yes	Yes	Yes	Yes	Yes	Yes
Veber	Yes	Yes	Yes	Yes	Yes	Yes	Yes	Yes	Yes	Yes	Yes	Yes	Yes	Yes
Egan	Yes	Yes	Yes	Yes	Yes	Yes	Yes	Yes	Yes	Yes	Yes	Yes	Yes	Yes
Muegge	No	Yes	Yes	No	Yes	Yes	Yes	Yes	Yes	Yes	Yes	Yes	Yes	No
Bioavailability Score	0.55	0.85	0.55	0.55	0.56	0.55	0.55	0.55	0.55	0.55	0.55	0.55	0.55	0.56

**Table 6 metabolites-13-00064-t006:** Binding affinity, conventional hydrogen-bonding, the number of closest interacting residues and distance to closest interacting residue (Å) of some compounds with best scores with the different targeted receptors (1HD2, 2XCT, 2QZW, and 3LN1).

Complexes	Binding Affinity (kcalxmol^−1^)	Conventional H-Bonds	No. Closest Interacting Residues	Closest Interacting Residue	
				Residue	Distance (Å)
1HD2/2-Phenylaminoadenosine	−6.7	4	7	Arg86	1.822
1HD2/Atractyloside	−6.8	8	8	Gly92	2.153
2XCT/10-Hydroxyloganin	−7.6	6	4	Asp1105	1.904
2XCT/Harderoporphyrin	−8.2	6	5	Arg1377	2.186
2QZW/2-Phenylaminoadenosine	−8.3	6	9	Arg195	2.223
2QZW/Atractyloside	−8.3	6	7	Glu132	2.023
3LN1/Harderoporphyrin	−9.9	6	7	Ser160	1.898
3LN1/Dihydro-Obliquin	−9.2	1	9	Ser516	2.497

**Table 7 metabolites-13-00064-t007:** Comparison between *D. flabellifolia* methanolic and methanol/water extract antioxidant activities.

*D. flabellifolia*/Tests	DPPH IC_50_ (mg/mL)	ABTS IC_50_ (mg/mL)	β-Carotene IC_50_ (mg/mL)
Methanolic extract	0.05 ± 0	0.105 ± 0	5.00 ± 0.78
Methanol/Water extract *	0.014 ± 0.045	0.102 ± 0.024	7.80 ± 0.919

* Snoussi et al. [5].

## Data Availability

Not applicable.

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
