# Peer review of "Chemical Composition of *Ducrosia flabellifolia* L. Methanolic Extract and Volatile Oil: ADME Properties, In Vitro and In Silico Screening of Antimicrobial, Antioxidant and Anticancer Activities"

_metabolites, 2022, doi:10.3390/metabo13010064_

Round 1
Reviewer 1 Report
The manuscript by Snoussi et al. describes isolation of constituents from Ducrosia flabellifolia L., their identification and study of biological activity. The in silico methodologies were also applied to rationalize the experimental data. The antimicrobial, antioxidant, and anticancer assays were applied. The results are important and deserve publication. However, the manuscript suffers from numerous flaws listed below.
Line 108: It is not clear how the authors selected the biotargets 1HD2, 2XCT, 2QZW, and 3LN1 for the docking study. Please explain here of below in the text of the manuscript.
Section 2.2.2: Please give more details in the description of HR-LCMS experiment: ionization mode(s), etc.
Section 2.6.1: A more detailed description of ADME properties determination (calculation) is necessary.
Line 185: The type of receptor is not indicated in parentheses for 3LN1.
Section 2.6.2: In the description of docking calculations, important information is omitted. Thus, the protein preparation options should be clarified. Also, the used options of AutoDock Vina program could be interesting to a reader. The name of the program "AutoDock Vina" should be mentioned properly along with the version of the software. Whether the receptors were treated as rigid or flexible?
Line 202: From the sentence, it is not clear whether this is the yield of essential oil or the yield of extract. Please modify the sentence.
Line 203: The value and its error are given with different number of decimal places. The same refers to lines 213 and 324.
Line 212: The phrase "yield of extraction" is ambiguous. Please check also throughout the manuscript.
Line 247: The phrase "spectrometric technique" is too common.
Lines 248,-250: Too many digits in the errors. One or two significant digits are desirable (e.g., errors 1.7 instead of 1.633, 0.49 instead of 0.482, and 0.70 instead of 0.694; the errors are rounded up). The TPC, TFC, and TTC magnitudes should be rounded correspondingly with normal rounding rules. The same refers to lines 325, 347, 353-355 (except the last value in Line 355), and the last column of Table 5.
Table 4: The notations like "0.023 ± 3x10-4" look strange. The same refers to lines 262-263.
Line 269: Too many significant digits in the IC50 values. Obviously, such a precision is unattainable in biological experiments. The same refers to lines 398-399,
Please check precisions and number of significant digits carefully also throughout the manuscript.
Table 5: The "cm/s" unit for Log Kp is very strange.
Figure 4: The legends on the panels are of low resolution.
Line 416: The PDB code should be given in capital: 3LN1.
Line 451: The conjunction "and" should be written in low case letters. Please check also throughout the manuscript.
Lines 460-463: Funding sources are not indicated.
The file with supplementary materials should also contain the title of the manuscript, author names and affiliations.
Abnormal self-citation was detected. For example, 13 referenced papers were co-authored by A. Kadri.
Summarizing, I recommend major revision of the manuscript before publication.
Reviewer 2 Report
In the manuscript entitled “Chemical Composition of Ducrosia flabellifolia L. Methanolic Extract and Volatile Oil: ADME Properties, In Vitro and In Silico Screening of Antimicrobial, Antioxidant and Anticancer Activities”, Authors described a potential antimicrobical, antioxidant and anticancer activity of a Ducrosia flabellifolia L. Methanolic Extract. The topic would be of interest and worth to be furthered, however, manuscript presents some critical issues regarding the results presented. List of criticisms are described below:
Major critical points:
1)Regarding antitumoral activity evaluation, no analysis was performed on non-tumoral cells. Isn't it conceivable that a methanolic extract rich of different types of compound can alter the vitality of any type of cell line? No indication was provided
2)Regarding the antimicrobical activity, has the methanol (or DMSO) effect been evaluated? No indication was provided.
3)The discussion session lacks of many comments regarding the results obtained by authors (see below).
Others points of criticism are listed below:
-What proof does the evaluation of the essential oil and the methanolic extract yields in percentage terms give to us? What is the term of comparison?
-No information about the tripeptides found in the methanolic extract were given
T -The low antioxidant activity on β-carotene compared to DPPH and ABTS also compared to the standard antioxidants used information was not discussed by authors.
-Figure 2 legend: do error bars indicate SEM or SDs?
-Regarding ADME predictions, the fact that most of the selected compounds are not inhibitors of the cytochrome P450 isoenzymes was not discussed by authors.
· -Figure 3 is not easily readable; in addition, the fact that the most of the selected compounds are not (or not fully? Is not clear) ideal for oral bioavailability was not discussed by authors.
- -The result regarding the interaction between Harderoporphyrin and 2QZW is nor showed.
- -The discussion session is too informal, especially in the first paragraph.
-What are the terms of comparison between the twenty-two phytochemical compounds identified by authors and the twenty-three bioactive compounds described in the discussion session (referred to Snoussi et al.)? Is not clear at all.
Round 2
Reviewer 1 Report
The authors improved the manuscript significantly, and now it can be accepted for publication after minor corrections.
Please replace the word "smiley" with the abbreviation "SMILES" in Line 426.
Also, the current version of AutoDock Vina program is v1.2.3. The reported version 1.5.6 (Line 438) obviously refers to AutoDock Tools, not Vina. Please correct.
Author Response
Review Report Form
Open Review
(x) I would not like to sign my review report
( ) I would like to sign my review report
English language and style
( ) English very difficult to understand/incomprehensible
( ) Extensive editing of English language and style required
( ) Moderate English changes required
(x) English language and style are fine/minor spell check required
( ) I don't feel qualified to judge about the English language and style
|
Yes |
Can be improved |
Must be improved |
Not applicable |
|
|
Does the introduction provide sufficient background and include all relevant references? |
(x) |
( ) |
( ) |
( ) |
|
Are all the cited references relevant to the research? |
(x) |
( ) |
( ) |
( ) |
|
Is the research design appropriate? |
(x) |
( ) |
( ) |
( ) |
|
Are the methods adequately described? |
( ) |
(x) |
( ) |
( ) |
|
Are the results clearly presented? |
(x) |
( ) |
( ) |
( ) |
|
Are the conclusions supported by the results? |
(x) |
( ) |
( ) |
( ) |
Comments and Suggestions for Authors
The authors improved the manuscript significantly, and now it can be accepted for publication after minor corrections.
Please replace the word "smiley" with the abbreviation "SMILES" in Line 426.
Response: Special thanks for the comment. Modified as recommended.
Also, the current version of AutoDock Vina program is v1.2.3. The reported version 1.5.6 (Line 438) obviously refers to AutoDock Tools, not Vina. Please correct.
Response: Special thanks for the comment. Modified as recommended.
Submission Date
29 November 2022
Date of this review
23 Dec 2022 16:37:04

Reviewer 2 Report
Authors have substantially improved the quality of their manuscript;
The resolution of figure 2 could be better refined.
Author Response
Review Report Form
Open Review
(x) I would not like to sign my review report
( ) I would like to sign my review report
English language and style
( ) English very difficult to understand/incomprehensible
( ) Extensive editing of English language and style required
( ) Moderate English changes required
(x) English language and style are fine/minor spell check required
( ) I don't feel qualified to judge about the English language and style
|
Yes |
Can be improved |
Must be improved |
Not applicable |
|
|
Does the introduction provide sufficient background and include all relevant references? |
(x) |
( ) |
( ) |
( ) |
|
Are all the cited references relevant to the research? |
(x) |
( ) |
( ) |
( ) |
|
Is the research design appropriate? |
(x) |
( ) |
( ) |
( ) |
|
Are the methods adequately described? |
(x) |
( ) |
( ) |
( ) |
|
Are the results clearly presented? |
( ) |
(x) |
( ) |
( ) |
|
Are the conclusions supported by the results? |
(x) |
( ) |
( ) |
( ) |
Comments and Suggestions for Authors
Authors have substantially improved the quality of their manuscript;
The resolution of figure 2 could be better refined.
Response: Special thanks for the comment. Modified as recommended.
Submission Date
29 November 2022
Date of this review
22 Dec 2022 17:54:34
